# Representativeness of Two Global Gridded Precipitation Data Sets in the Intensity of Surface Short-Term Precipitation over China

**Xiaocheng Wei [1,2], Yu Yu [3,]*[ID], Bo Li [1,2] and Zijing Liu [4]**

1   Key Laboratory of Radiometric Calibration and Validation for Environmental Satellites (LRCVES/CMA), National Satellite Meteorological Center, China Meteorological Administration (NSMC/CMA), Beijing 100081, China
2   Innovation Center for FengYun Meteorological Satellite (FYSIC), Beijing 100081, China
3   National Meteorological Information Centre, China Meteorological Administration, Beijing 100081, China
4   Guangzhou Meteorological Satellite Ground Station, Guangzhou 510630, China
*   Correspondence: yuyu@cma.gov.cn

**Abstract:** This study evaluates the representativeness of two widely used next-generation global satellite precipitation estimates data for short-term precipitation over China, namely the satellite data from the Climate Prediction Center morphing (CMORPH) and the satellite data from the Global Precipitation Measurement (GPM) mission. These two satellite precipitation data sets were compared with the hourly liquid *in-situ* precipitation from China national surface stations from 2016 to 2020. The results showed that the GPM precipitation data has better representativeness of surface short-term precipitation than that of the CMORPH data, and these two quantitative precipitation estimate (QPE) data sets underestimated extreme precipitation. Moreover, we analyzed the influence of the error between two QPE data sets and the *in-situ* precipitation on the classification of short-term precipitation intensity. China uses 8.1–16 mm/h as the definition of heavy precipitation, but the accuracy of the satellite QPE product was different due to the different lowest threshold of heavy rain (more than 8.1 mm/h or more than 16 mm/h). Increasing the threshold value of the QPE data for short-term strong precipitation resulted in lower accuracy for detecting such events, but higher accuracy for detecting moderate intensity rainfall. When studying short-term strong precipitation over China using precipitation grade, selecting an appropriate threshold was important to ensure accurate judgments. Additionally, it is important to account for errors caused by QPE data, which can significantly affect the accuracy of precipitation grading.

**Keywords:** precipitation intensity; GPM; CMORPH; rain gauge

## 1. Introduction

Extreme precipitation events have a great influence on human life and property [1], so high-resolution precipitation observations are essential for mitigating disasters [2]. Generally, timely and unbiased precipitation analysis builds the basis of hydro-meteorological forecast and can help us further understand and manage agricultural and hydrologic risks [3–6]. To meet the great demand for high-quality rain-gauge-based global precipitation analysis data sets to serve as *in-situ*-based references, great efforts have been made since 1989 at the Global Precipitation Climatology Centre (GPCC) of Deutscher Wetterdienst mandated by the World Meteorological Organization World Climate Research Program. Several upgraded editions of the land surface precipitation analysis have been continuously published since the first one [7], expanding the spatial–temporal coverage and product categories [8,9]. The National Centers for Environmental Prediction (NCEP) of the U.S. National Oceanic and Atmospheric Administration (NOAA) [10] and the Climate Research Unit of the University of East Anglia [11] have also produced related works.

However, the density of ground-based stations is not even, and there are even no measurements on some complex terrain areas [12], such as mountainous regions, oceans, deserts and sparsely populated areas. Thereby, the absolute precision of the rain-gauge-based analysis is mainly affected by both the density and configuration of the rain gauge network, as well as the interpolation methods [13,14]. With the rapid development of instruments on remote sensing satellites, such as microwave sensors onboard polar-orbiting satellites and the infrared (IR) sensors onboard geostationary satellites [15,16], new global and regional precipitation estimate data sets are now available [17–19]. As an efficient and reliable way to retrieve high spatial–temporal resolution precipitation, the remote sensing satellite precipitation products not only make up for the shortcomings of uneven distribution and difficult maintenance of ground-based rain gauge network, but also avoid the interference of ground-based radar signals (e.g., high mountains, towers, etc.) [20,21].

We utilized the Global Precipitation Measurement (GPM) satellites data and the NOAA Climate Prediction Center (CPC) Morphing Technique (CMORPH) data, which are the two most widely used quantitative precipitation estimate (QPE, unit: mm/h) data with a spatial resolution of half an hour and good performance in previous studies on short-term precipitation over China [22–24]. The NASA (National Aeronautics and Space Administration) GPM satellite mission as the successor of the Tropical Rainfall Measurement Mission (TRMM) [25,26], not only inherits its basic ability in observing large- and medium-sized tropical and subtropical precipitation, but can also detect frozen precipitation and slight precipitation (<0.5 mm/h) with higher accuracy by carrying more advanced dual-frequency precision radar (DPR) and the passive microwave sensor (GPM Microwave Imager, GMI) [27]. Based on these satellite precipitation measurement missions, a merged precipitation product with a time interval of 30 min and a horizontal spatial resolution of $0.1° \times 0.1°$ is routinely generated as a typical level-3 gridded Integrated Multi-Satellite Retrievals for GPM (GPM IMERG). This gridded product collocates, inter-calibrates, merges and interpolates some satellite microwave and microwave-calibrated satellite IR-based precipitation estimates, ground-based rain gauge data, and other precipitation estimates on a global scale. Similarly, the CMORPH data also produces a global precipitation analysis with the same temporal resolution as GPM (30 min temporal resolution) and higher spatial resolution than GPM ($0.07277° \times 0.07277°$) [28]. These two QPE databases are widely used in many different aspects because they can obtain the precipitation intensity and coverage reasonably well.

Some former studies have assessed the applicability of these two QPE data sets. The CMORPH and GPM data sets over China have also been used for hydro-meteorological and climate studies in China [29–32]. When Tian et al. [33] evaluated some precipitation products against ground-based rain gauge only and the Doppler radar measurement data corrected by the rain gauge data, and they found that the CMORPH data set showed the obvious season-dependent biases, with overestimation in summer and underestimation in winter. Their study also demonstrated that the CMORPH data set had an insignificantly lower uncertainty and a higher probability of detection of rain events at shorter time scales than others. Yang et al. [34] appraised the reliability of four satellite estimate precipitation data sets in the arid regions of Northwest China by comparing them with ground-based or reported values on daily scale from 2003 to 2010. Yang et al. [34] also found that satellite estimate precipitation products were more accurate in the warm season than in the cold season, and that the CMORPH data set tended to overestimate precipitation. For the GPM, Nan et al. [35] used eight independent statistical and detection indicators to assess the performance of the GPM IMERG precipitation products in China. Their results showed a good satellite precipitation-detection ability in southeastern China, with the related root-mean-square error (RMSE) increasing from northwest to southeast. Fang et al. [36] also evaluated the performances of the IMERG data in extreme precipitation estimation over China, found that the quality of the GPM IMERG data with extreme rainfall rate was restricted, and the data quality was affected by the underestimation of the extreme precipitation.

Therefore, it is not difficult to see that these two QPE products have different degrees of defects in describing surface precipitation. However, in many current studies (especially in precipitation studies using machine learning), the estimated precipitation of the QPE product is often graded, and the precipitation intensity grade is used as the label for further research. This led us to think about the representativeness of the graded QPE products on surface precipitation intensity. For this issue, by using high-quality ground-based rain gauge data in China, it should comprehensively evaluate the qualities and representativeness of two global gridded quantitative precipitation estimate data sets mentioned above. In particular, another critical issue is that if these two independent satellite QPE products can be used as the quantitative criteria or indicators for determining the discrete intensity of heavy precipitation or not. At present, the existing research is more inclined to analyze the error of satellite retrieval precipitation data over different places, and more inclined to analyze the representativeness of daily precipitation rather than hourly precipitation [23,37,38]. Furthermore, no research has focused on the representativeness of QPE hourly precipitation data on precipitation intensity grade. For short-term strong precipitation, it is obviously unable to meet the requirements if we only analyze the representativeness of QPE daily precipitation in distinguishing precipitation intensity grades. In this study, the matched precipitation samples were categorized into three piecewise indicators according to the commonly used classification of precipitation intensity in China: >0 mm/h and <2.5 mm/h for light rain, 2.5–16 mm/h for moderate rain and heavy rain, and >16 mm/h for rainstorms (8 mm/h is often used as the boundary to distinguish between heavy rain and rainstorms). Meanwhile, in the study of extreme precipitation events, the minimum precipitation threshold that everyone pays attention to is often different, and the judgment accuracy of the satellite estimate QPE data at different minimum precipitation thresholds is also different. The primary purpose of this exploration therefore is to assess both the quantitative characteristics of these two independent gridded satellite QPE data and their applicability as a quantitative indicator for estimating precipitation intensity (especially heavy rain) over China.

The rest of this paper is arranged as follows. Section 2 briefly introduces the two satellite estimated QPE data sets and the ground-based rain gauge data in China, as well as the intuitive error indicators for evaluating satellite QPE data. Section 3 presents the primary results and discussions. Section 4 summarizes the major conclusions and emphasizes the findings of this investigation.

## 2. Data and Methods

The data we used in this study contained the hourly *in-situ* precipitation data in China and two satellite precipitation estimate data sets from the CMORPH and the GPM, spanning the years 2016–2020. In order to make the accuracy of the evaluation of QPE data in estimating the precipitation intensity grade more convincing, we will first analyze the errors between these two gridded satellite QPE data sets and the *in-situ* precipitation over China, and then analyze the accuracy of QPE data in estimating the precipitation intensity grade based on the results of error analysis.

The China hourly *in-situ* precipitation data were derived from 2167 national ground-based rain gauge stations, which are archived in the National Meteorological Information Centre (NMIC), China Meteorological Administration (CMA). The geographical locations of these stations are shown in Figure 1. The evaluation of the rain gauge measurements indicates that the data availability was higher than 95% after undergoing the quality control procedures, namely checking the national climatological thresholds, checking the regional climatological thresholds, checking temporal consistency and checking spatial consistency [39]. These ground measurements have already been employed to assess the quality of some of the satellite estimate QPE data, such as the TRMM precipitation product 3B42 [40], and to further manifest the suitability of the GPCC Full Data Daily Analysis from a climatic perspective [41].

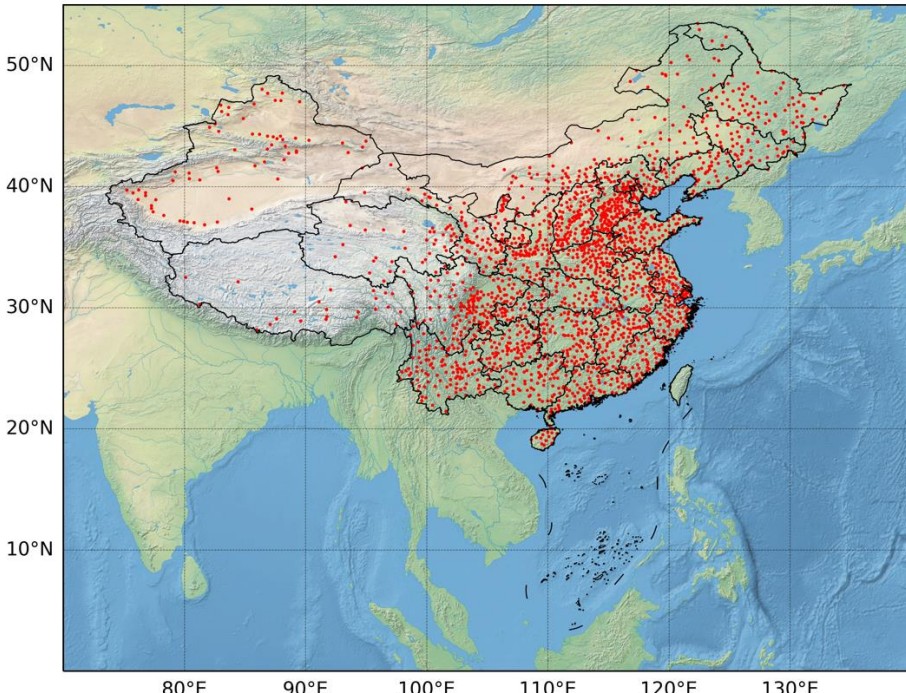

**Figure 1.** Locations (red solid circle) of the 2167 national rain gauge stations in China.

Taking the *in-situ* precipitation as the real value or truth, this study analyzed the error of two widely used hourly satellite QPE data sets. Currently, CMORPH is the global precipitation product with the finest spatial resolution. It relies exclusively on passive microwave and geostationary infrared band radiance observations, with no ground-based rain gauge information [28]. The data version used in this investigation is V1.0, within a 30 min temporal resolution and a horizontal spatial resolution of 0.07277° × 0.07277° (about 8 km × 8 km at the equator).

The GPM mission as an international network of satellites that can implement next-generation global observations on liquid and solid precipitation (or snow), supported by the U.S. NASA and the JAXA (Japan Aerospace Exploration Agency) [42]. The GPM Core Observatory was successfully launched on February 27 of 2014. It is regarded as an extension of the prior TRMM mission, which focused primarily on heavy to moderate rain over tropical and subtropical ocean areas. It carries the world's first space-borne Ku/Ka-band DPR and a multi-channel GMI. The DPR instrument is able to measure three-dimensional precipitation structures over 78 and 152 miles (125 and 245 km) swaths, which consists of a Ka-band radar (KaPR at 35.5 GHz) and a Ku-band radar (KuPR at 13.6 GHz) [35]. The data used in this study is the GPM IMERG Final Precipitation Level-3 data (GPM-3IMERGHH), with a 30 min temporal resolution and a 0.1° × 0.1° horizontal spatial resolution. As described on the GPM-3IMERGHH product release website, the input precipitation estimates computed from the various satellite passive microwave sensors are intercalibrated to the Combined Ku Radar-Radiometer Algorithm (CORRA) product, then "forward/backward morphed" and combined with microwave precipitation-calibrated geo-IR fields, and finally adjusted with seasonal Global Precipitation Climatology Project (GPCP) Satellite Gauge (SG) surface precipitation data to provide GPM-3IMERGHH data. The data type of the last step is not used when generating CMORPH data.

It is imperative to note that when matching QPE products and *in-situ* data, the satellite estimate QPE product with the national surface stations as the center and 1° as the radius is interpolated continuously in the north and south directions. In this way, the value of satellite QPE data at the location of the national surface stations can be obtained by matching the longitude and latitude.

Figure 2 shows the comparisons of precipitation horizontal distribution between gridded CMORPH and GPM data at 08:00 UTC on 8 August 2018 and at 18:00 UTC on 6 June 2019, over China and its surrounding areas. As can be seen, these two satellite QPE data were relatively consistent in the location and intensity of heavy precipitation. However, when comparing Figure 2a with Figure 2b (or comparing Figure 2c with Figure 2d), it is not difficult to see that GPM was more sensitive to clouds with relatively light precipitation (≤1 mm/h), which makes it perform better than the CMORPH data in the continuity of precipitation area.

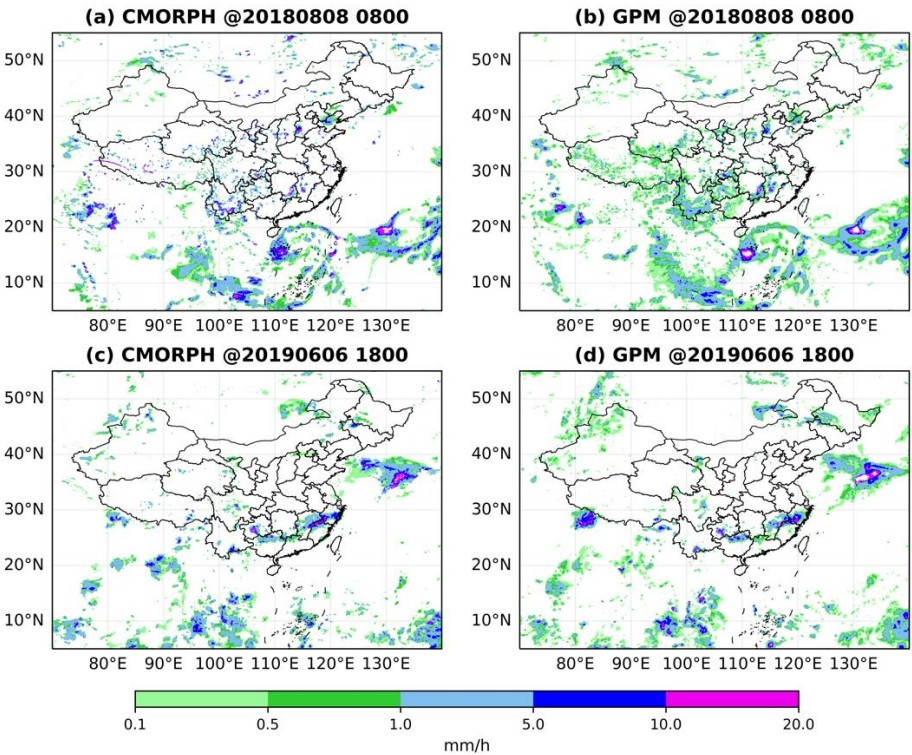

**Figure 2.** Comparisons of precipitation horizontal distribution between the (**a,c**) CMORPH data and (**b,d**) GPM data at 08:00 UTC on 8 August 2018 (top panel) and at 18:00 UTC on 6 June 2019 (bottom panel) over China and its surrounding areas.

Note that when calculating the error indicators, we did not consider or screen out the data if the national surface station had fewer than five samples. At the same time, in order to highlight the distribution characteristics of errors, we only discuss the cases where the satellite QPE data and the *in-situ* precipitation data were both greater than 0 mm/h. In addition, this analysis only considered the issue of liquid precipitation, which excludes QPE data influenced by ice and snow [43].

A few intuitive error indicators, namely mean absolute error (MAE), mean bias error (MBE), root mean square error (RMSE) and correlation coefficient (R), were used to evaluate the error between *in-situ* observed precipitation and the satellite QPE data. These indicators are calculated by Equations (1)–(4), in which n is the sample size, $y_{in\text{-}situ}$ is the *in-situ* precipitation and $y_{sat}$ is the satellite QPE data. The definitions of $Cov(y_{in\text{-}situ}, y_{sat})$ and Var are shown in Equations (5) and (6).

$$\mathbf{MAE} = \frac{1}{\mathbf{n}} \sum_{\mathbf{i}=1}^{\mathbf{n}} \left| \mathbf{y_{sat,i}} - \mathbf{y}_{in-situ,\ i} \right| \tag{1}$$

$$\mathbf{MBE} = \frac{1}{\mathbf{n}} \sum_{\mathbf{i}=1}^{\mathbf{n}} \left( \mathbf{y_{sat,i}} - \mathbf{y}_{in-situ,\ i} \right) \tag{2}$$

$$\mathbf{RMSE} = \sqrt{\frac{1}{\mathbf{n}}\sum_{\mathbf{i=1}}^{\mathbf{n}}\left(\mathbf{y_{sat,i}} - \mathbf{y}_{in-situ,\ \mathbf{i}}\right)^2} \tag{3}$$

$$\mathbf{R} = \frac{\mathbf{Cov}\left(\mathbf{y}_{in-situ\prime}\ \mathbf{y_{sat}}\right)}{\sqrt{\mathbf{Var}(\mathbf{y}_{in-situ})\mathbf{Var}(\mathbf{y_{sat}})}} \tag{4}$$

$$\mathbf{Cov}\left(\mathbf{y}_{in-situ\prime}\ \mathbf{y_{sat}}\right) = \frac{\sum_{\mathbf{i=1}}^{\mathbf{n}}\left(\mathbf{y}_{in-situ,\ \mathbf{i}} - \overline{\mathbf{y}_{in-situ}}\right)\left(\mathbf{y_{sat,\ i}} - \overline{\mathbf{y_{sat,i}}}\right)}{\mathbf{n-1}} \tag{5}$$

$$\mathbf{Var}\ (\mathbf{y}) = \frac{\sum_{\mathbf{i=1}}^{\mathbf{n}}(\mathbf{y_i} - \overline{\mathbf{y}})^2}{\mathbf{n-1}} \tag{6}$$

MAE is the mean absolute difference value between the *in-situ* precipitation and the satellite estimate QPE data. We also used RMSE, which is sensitive to outliers, while MAE is the opposite. Different from MAE, MBE considers both the absolute value and the sign of the error, which considers both positive and negative deviations. Moreover, the correlation coefficient R can roughly determine the linear correlation between various variables. The closer the absolute value of R is to 1, the higher the correlation between the QPE data and *in-situ* data.

## 3. Results and Discussions

### 3.1. Overall Error Distribution

Figure 3 shows the distributions of MAE, MBE, RMSE and R for the two satellite QPE products over China. As can be seen, the GPM data performed much better than the CMORPH data over China in error analysis and correlation with *in-situ* data. For CMORPH data, both the MAE and RMSE of CMORPH showed an increasing trend from northwest China to southeast China, and reached their peaks in the coastal areas of Hainan and Guangdong Province, which are in harmony with the results of Li et al. [44,45]. This north–south increasing trend connects well with distribution of precipitation intensity. The relatively heavy precipitation intensity over southeast China resulted in a large variation range of precipitation, and finally induced relatively large errors. In contrast, the stations with less precipitation all year round over northwest China produced relatively low errors.

Because the GPM data have been corrected by the seasonal GPCP SG surface precipitation data, the errors between the GPM data and *in-situ* precipitation were much smaller than that between the CMORPH data and *in-situ* precipitation. The MAE and MBE were all closer to 0 than the relevant error indicators of the CMORPH data, which indicates that the absolute error between the GPM data and *in-situ* data was smaller than the absolute error between the CMORPH data and *in-situ* data, and that for precipitation of all intensities, the QPE data often overestimates precipitation. As for the distribution of RMSE values, it can be seen from Figure 3e,f that the RMSE of Guangdong Province and Hainan Province was larger than that of other provinces in China (about 2 mm/h), but the overall RMSE value was smaller than that of CMORPH. From the distribution of R, it can be seen that the R value of the GPM data was higher than that of the CMORPH data as a whole, and the highest R value was more than 0.8. This proves that the GPM data and *in-situ* data had good correlation, which is more reliable at the site than the CMORPH data with the same interpolation processing in the process of matching data done by our research.

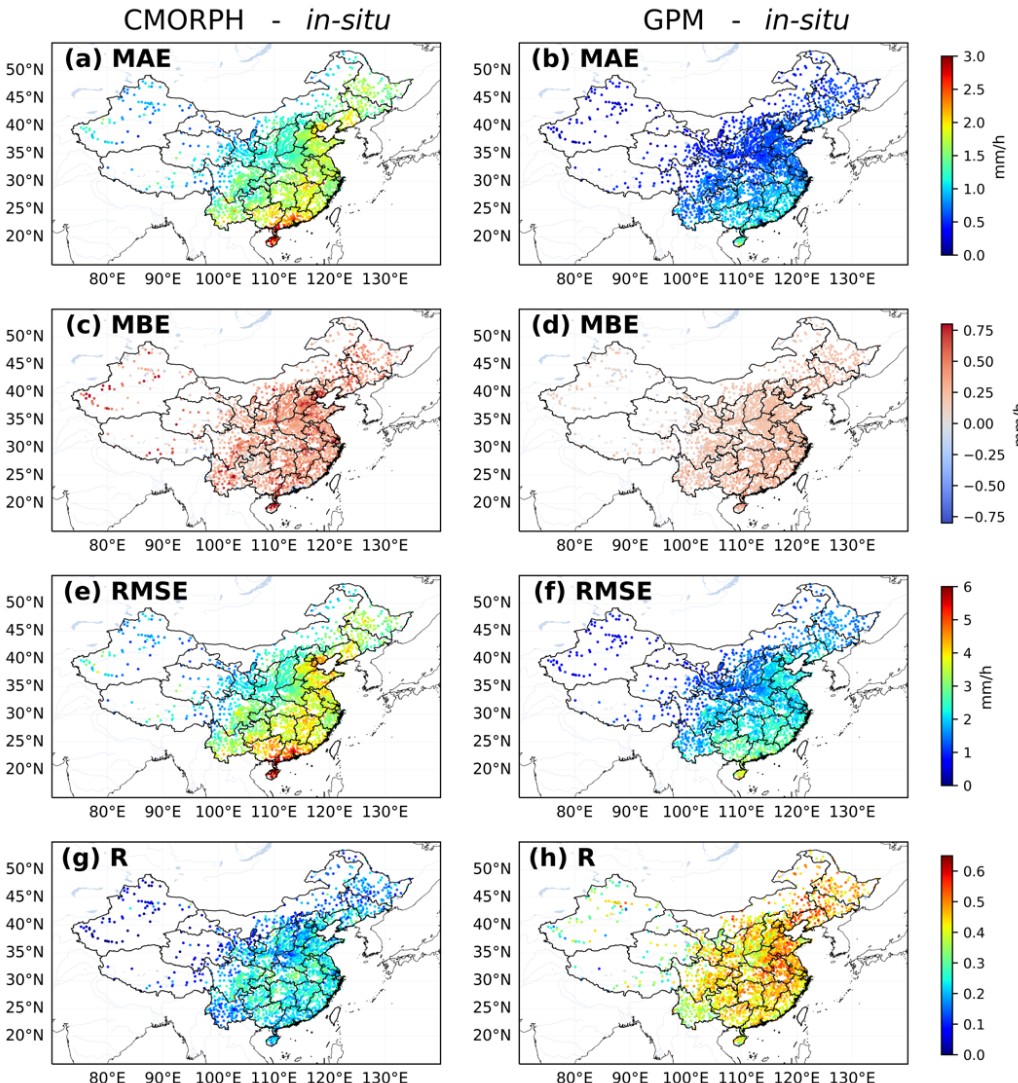

**Figure 3.** Distributions of four error indicators. Subfigures (**a**,**c**,**e**,**g**) show the distributions of MAE, MBE, RMSE and R of CMORPH data, respectively. The subfigures (**b**,**d**,**f**,**h**) represent the corresponding results of the GPM data.

### 3.2. Error Seasonal Distribution

Figures 4–7 show the seasonal distributions of MAE, MBE, RMSE and R of the two satellite QPE products. Following the division of seasons in China, we chose the following criteria for the division of seasons: March to May are spring, June to August are summer, September to November are autumn, and December, January and February are winter. Similar to the overall distribution, the seasonal performance of the GPM data was still better than that of the CMORPH data. However, both the CMORPH data and GPM data showed certain seasonal distribution characteristics and variations. Due to the hot and rainy weather in summer and the low temperature and drought in winter in China [46], the errors (MAE and RMSE) reached their maximums in summer and minimums in winter.

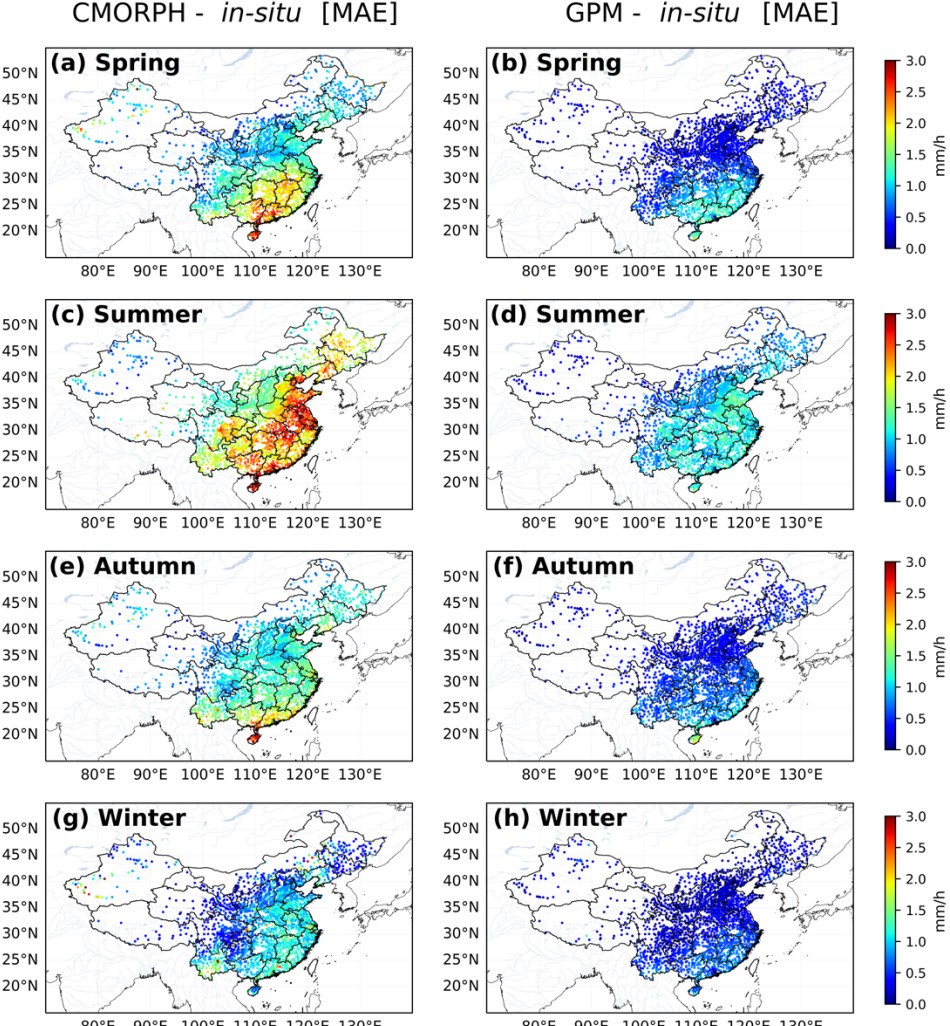

**Figure 4.** Distributions of the MAE in four seasons. Subfigures (**a**,**c**,**e**,**g**) show the MAE distributions of the CMORPH data. The subfigures (**b**,**d**,**f**,**h**) represent the corresponding results of the GPM data.

In spring, the high-value areas of MAE and RMSE of the GPM data and CMORPH data were observed in southern China, particularly in Guangdong Province and Hainan Province. The error between the GPM data and *in-situ* data was also more evenly distributed, and was less than the error between the CMORPH and *in-situ* data.

The error high value area between the QPE data and *in-situ* data has obviously became larger in summer. In addition to the high-error area that appeared in the spring, affected by the East Asian summer monsoon, the precipitation in eastern China increased, and the error between the CMORPH data and the *in-situ* data in these areas also increased. After entering autumn, these high-error areas gradually weakened and finally disappeared. Because we only calculated sites with valid data greater than 5, the data available for calculating R was significantly reduced in winter. Benefitting from the seasonal GPCP SG surface precipitation data correction, the R value of the GPM data and *in-situ* data were maintained at a high level, which also means that the GPM data and *in-situ* data had good correlation. However, in winter, there was a low-value area of R in northern China. Because this low-value area only appears in winter, it may be affected by reduced precipitation.

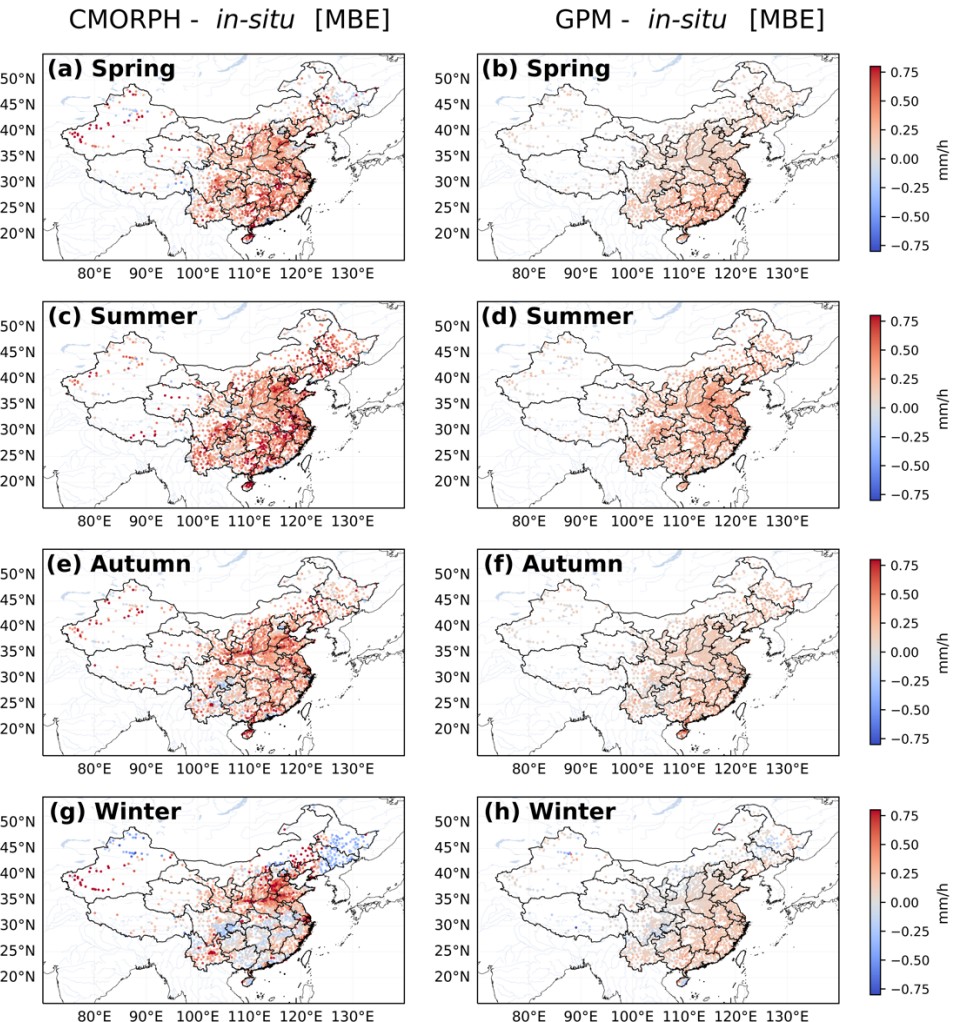

**Figure 5.** Same as Figure 4, but for the MBE.

### 3.3. Quality of Satellite Precipitation Estimate Products under Three Precipitation Grades

The errors of satellite QPE products under different precipitation grades have been studied in a previous study [44], but it focused on the daily precipitation estimated by the GPM or other satellites rather than the hourly precipitation of the CMORPH. However, the CMORPH and GPM data sets are extensively used in the study of convective precipitation, so the accuracy of hourly precipitation classification is much more important than the daily precipitation [47–50]. Moreover, although there is a certain error between the satellite QPE data and the *in-situ* precipitation [51], the impact of the error may not be as great as we think when we use precipitation to determine the convection intensity. When the criterion for judging the convection intensity is the precipitation intensity grade, rather than the absolute precipitation intensity, the influence of the error between the satellite QPE data and the *in-situ* precipitation will change, which is closely associated with the classification of the precipitation intensity grade. For example, when we divide the precipitation of 2.5–16 mm/h into the same intensity grade, no matter whether the actual precipitation is 8 mm/h or 10 mm/h, there will be no deviation in the judgment of intensity grade. Based on early research [29], we divided the hourly precipitation into three piecewise grades: 0–2.5 mm/h (excluding 0 mm/h), 2.5–16 mm/h and more than 16 mm/h, denoted as Grade–0, Grade–1 and Grade–2, respectively.

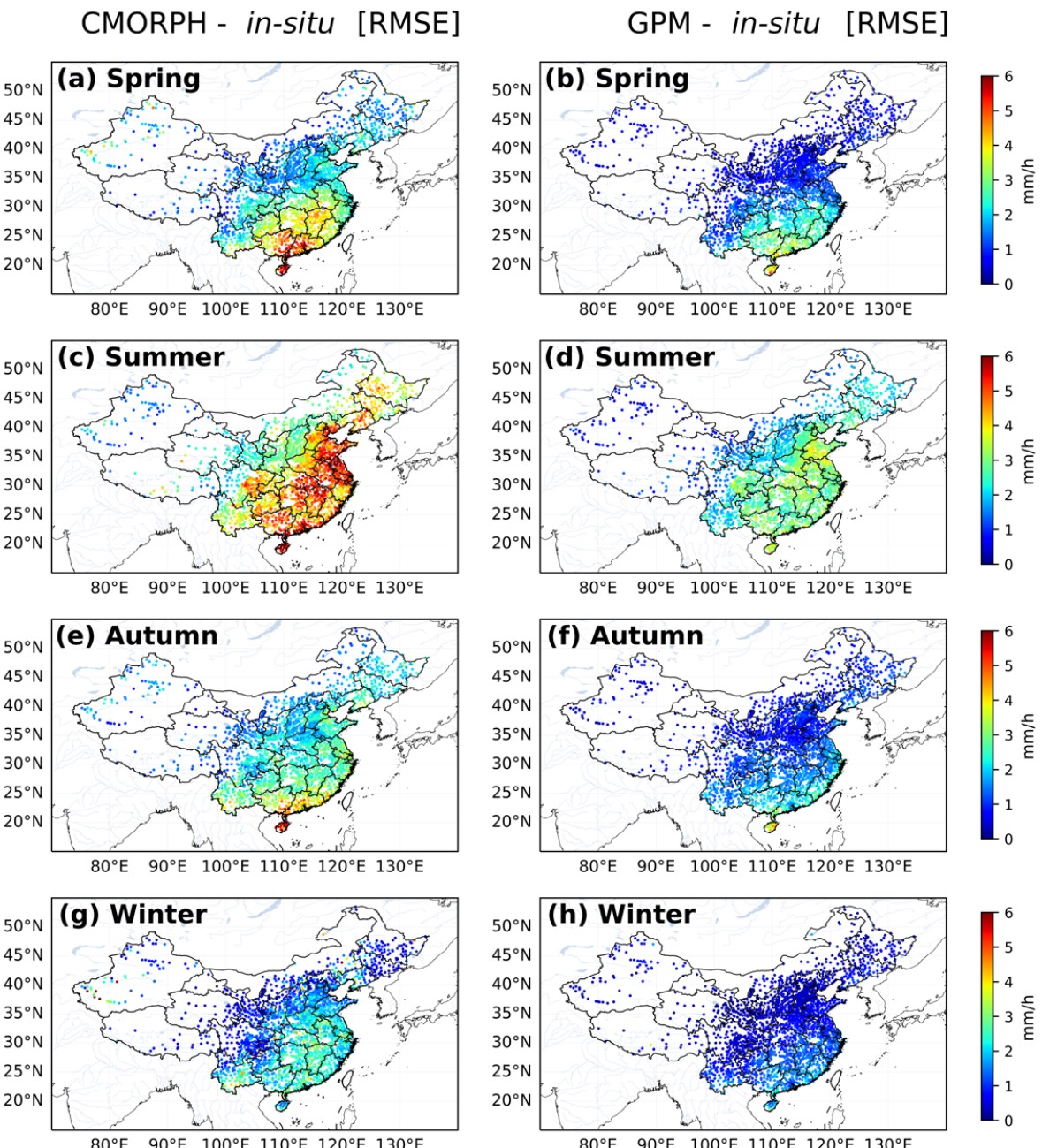

**Figure 6.** Same as Figure 4, but for the RMSE.

Furthermore, the probability of detection (POD), false alarm ratio (FAR) and critical success index (CSI) were used in this study to evaluate the performance of the satellite QPE data. These indices can be computed using the number of hits (H), false alarms (F) and misses (C) using Equations (7)–(9).

$$POD = \frac{H}{H + C} \tag{7}$$

$$FAR = \frac{F}{H + F} \tag{8}$$

$$CSI = \frac{H}{H + F + C} \tag{9}$$

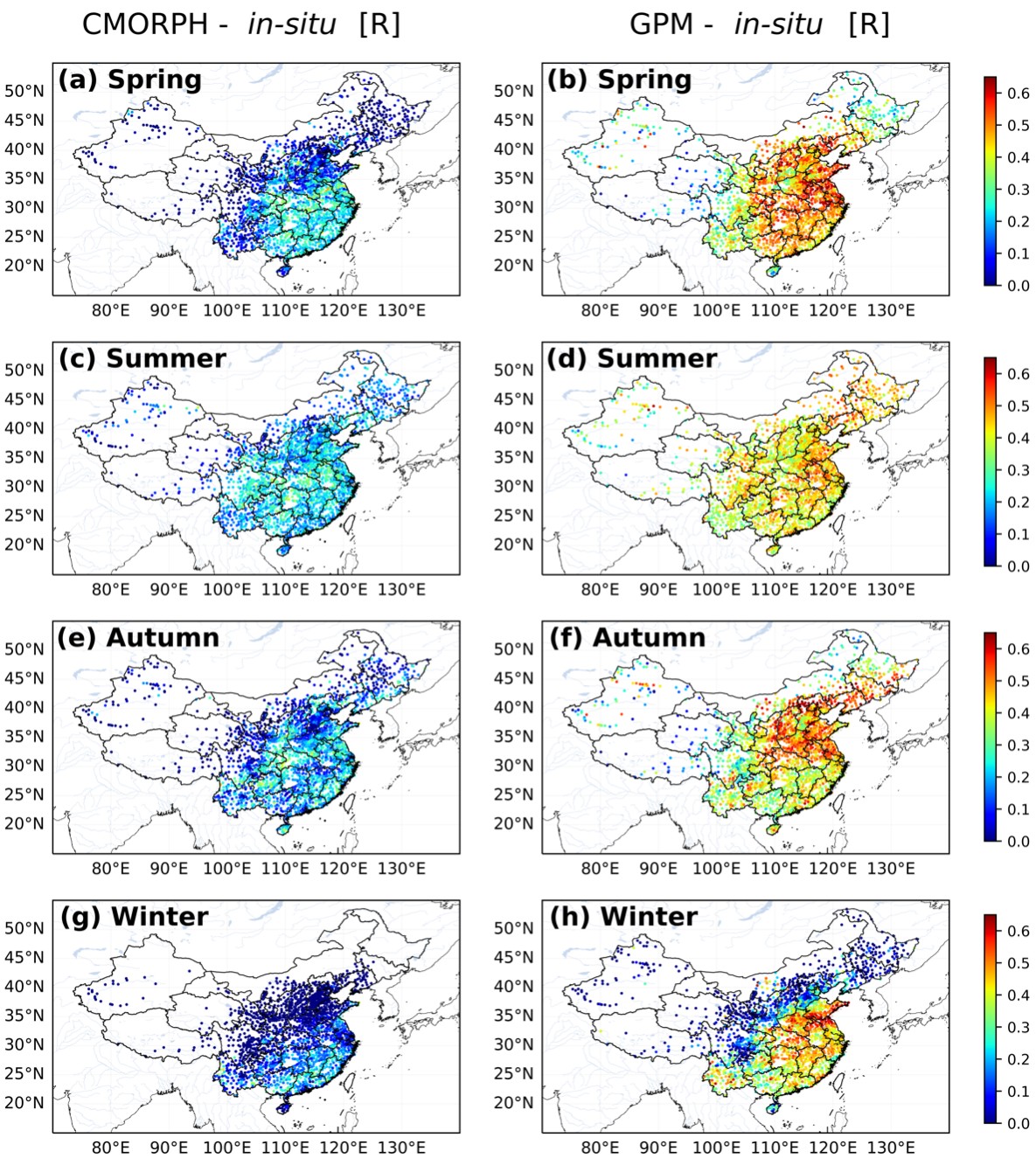

**Figure 7.** Same as Figure 4, but for the R.

Figures 8 and 9 show the POD, FAR and CSI of the three precipitation grades. The results show that both the GPM data with excellent performance in error analysis and the CMORPH data had similar performance in judging precipitation intensity grade. For Grade-0, the POD values were very high and close to 1, and the corresponding FAR was close to 0. With the increase in precipitation intensity, the POD and FAR values decreased and increased, respectively. For Grade-2, the POD and FAR values were close to 0.2 and 0.8, respectively. Affected by the low POD, the CSI scores representing the combined effect of POD and FAR were also very low. Because the definition of extreme precipitation is precipitation >16 mm/h and we have already carried out data quality control and eliminated the outliers, it can be inferred that the QPE data often seriously underestimate extreme precipitation. The statistical results also proved this point. The statistical results showed that, of the 71,931 extreme precipitation cases (*in-situ* precipitation intensity greater

than 16 mm/h) at 2167 stations from 2016 to 2020, the CMORPH data overestimated 2729 precipitation cases and underestimated 69,201 precipitation cases (96.2%), while the GPM data overestimated 3917 precipitation cases and underestimated 68,013 precipitation cases (94.6%). In order to show QPE's underestimation of extreme precipitation more intuitively, we selected the stations located in four regions that showed obvious errors in the above error analysis, and showed the extreme precipitation events that occurred at these four stations from 2016 to 2020. Figure 10 shows the observed Grade-2 (>16 mm/h) precipitation and the corresponding satellite QPE data at four different and typical ground-based stations, namely Hainan station (19.217°N, 110.481°E), Guangdong station (23.15°N, 113.017°E), Jiangsu station (31.4°N, 121°E) and Guizhou station (25.417°N, 107.883°E). It can be seen from the figure that, compared with the CMORPH data, the GPM data was closer to ground observation precipitation in some extreme precipitation cases, but there was still a big difference between the two QPE data sets and the *in-situ* data when extreme precipitation occurs. In general, the QPE data overestimated the light precipitation on the ground, but seriously underestimated the heavy precipitation.

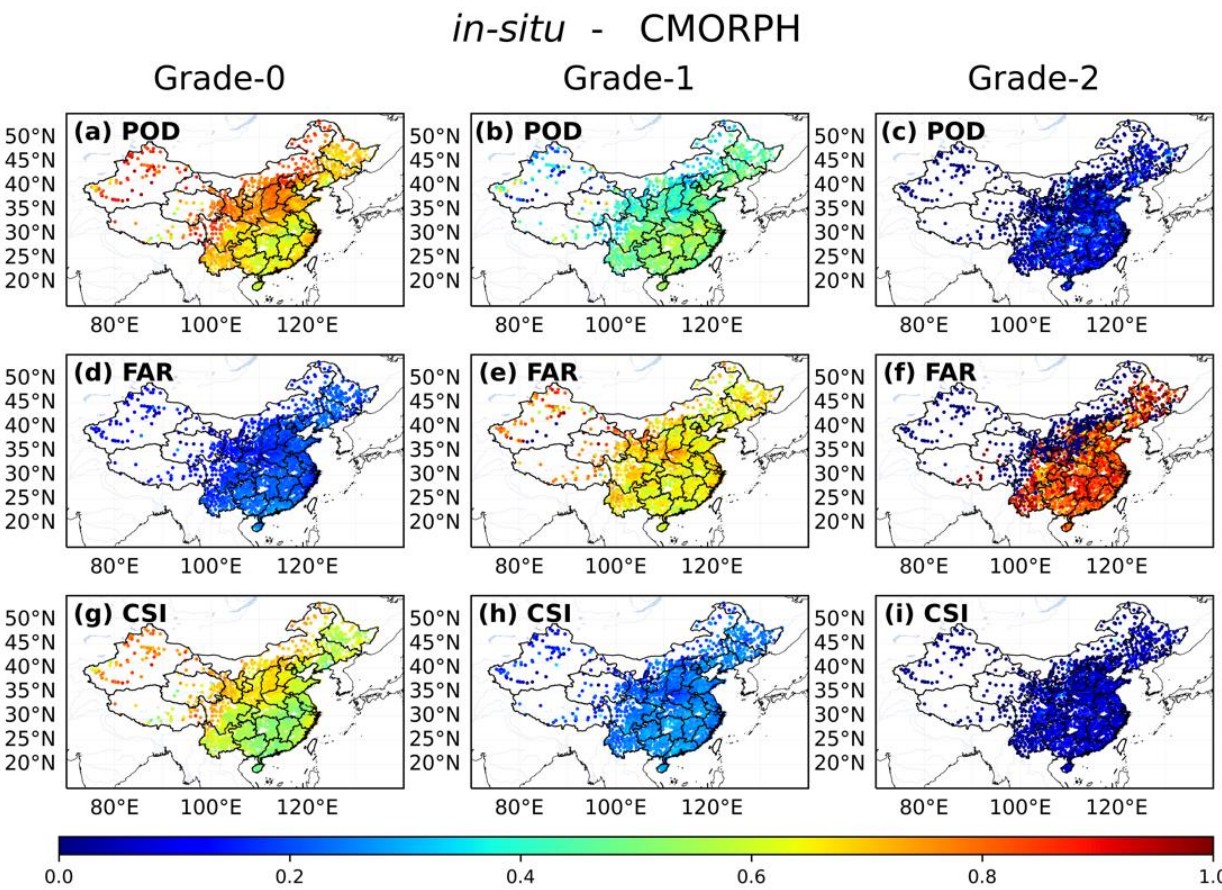

**Figure 8.** Distributions of the POD, FAR and CSI for the three piecewise grades of precipitation from CMORPH data. The first to third rows represent POD, FAR and CSI of different grades of CMORPH data, respectively, and the first to third columns represent the indices of Grade-0, Grade-1 and Grade-2, respectively.

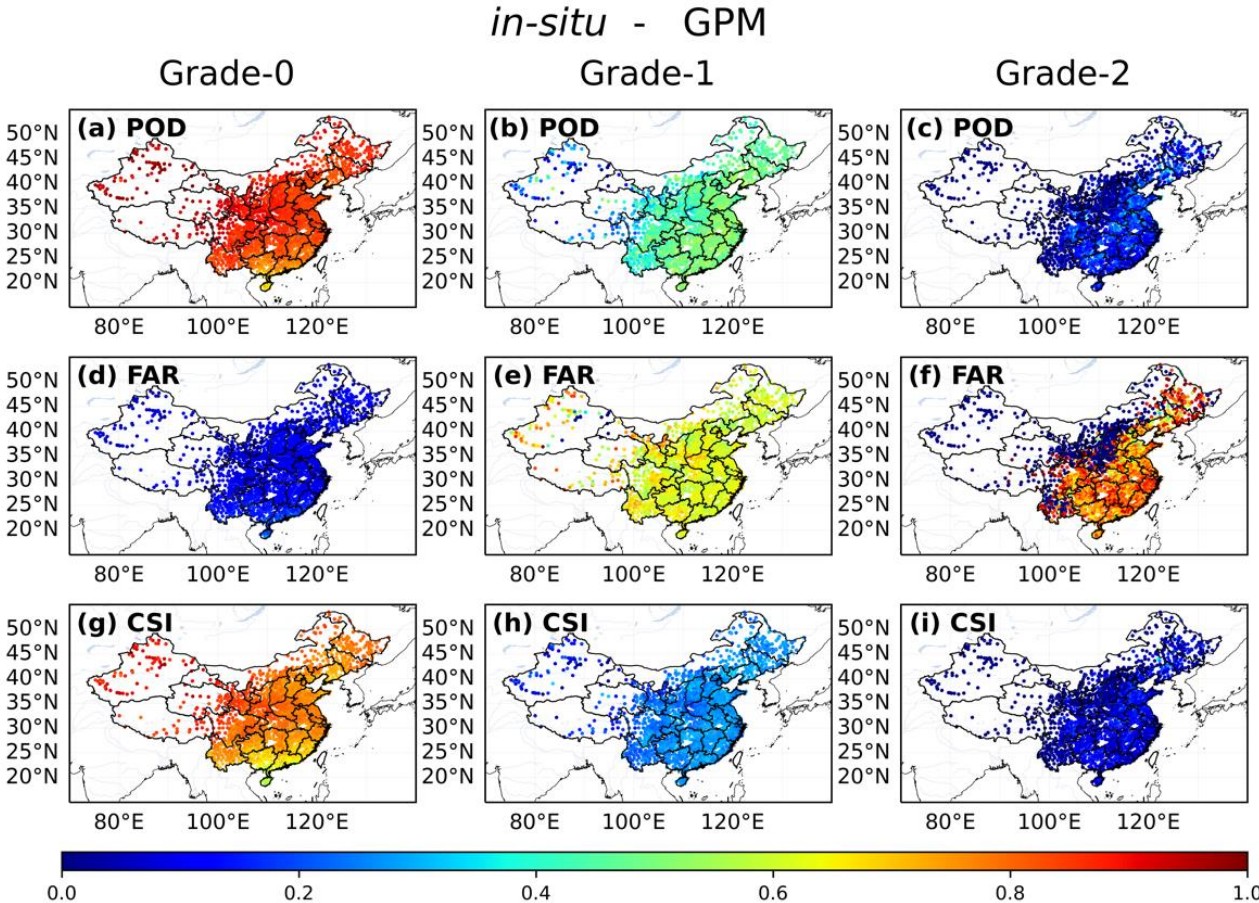

**Figure 9.** Same as Figure 8, but for GPM data.

In addition, we have also discussed the impact of modifying the classification thresholds on the classification accuracy of satellite QPE data. Figures 11 and 12 show the three indices of Grade-1 and Grade-2 under different classification thresholds. When the threshold of Grade-2 was low, the classification accuracy of Grade-2 became higher, and the classification accuracy of the corresponding Grade-1 became lower. When the threshold of Grade-2 increased, the results were the opposite. This finding profoundly reflects the serious underestimation of satellite QPE data on extreme precipitation.

In summary, QPE data (both GPM data with good performance in error analysis and CMORPH data with poor performance in error analysis) can represent weak or moderate precipitation on the ground well, but it cannot accurately represent the surface precipitation intensity in short-term extreme precipitation. QPE data was prone to judge strong precipitation as light precipitation, thus missing the cases of heavy precipitation. This incorrect judgment will directly have a negative impact on the research that uses QPE data to label precipitation intensity. When the threshold of defining extreme precipitation decreases, the accuracy of QPE data to judge extreme precipitation will increase, but the accuracy of judging moderate precipitation will decrease slightly. Research on extreme precipitation intensity based on QPE data needs to consider the error of QPE data.

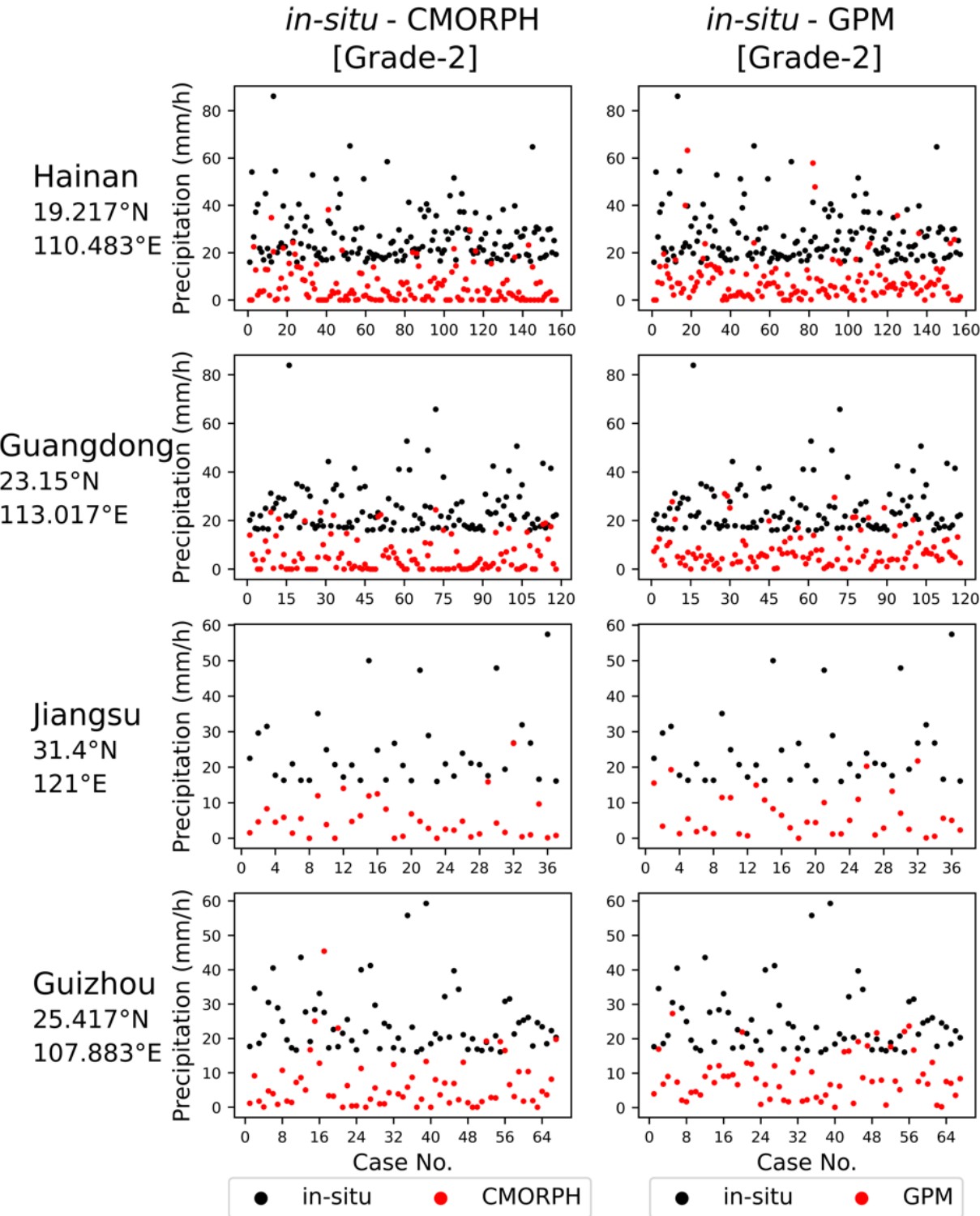

**Figure 10.** Observed Grade-2 precipitation and the corresponding satellite precipitation estimates in four different ground stations: Hainan station (first row: 19.217°N, 110.481°E), Guangdong station (second row: 23.15°N, 113.017°E), Jiangsu station (third row: 31.4°N, 121°E) and Guizhou station (fourth row: 25.417°N, 107.883°E).

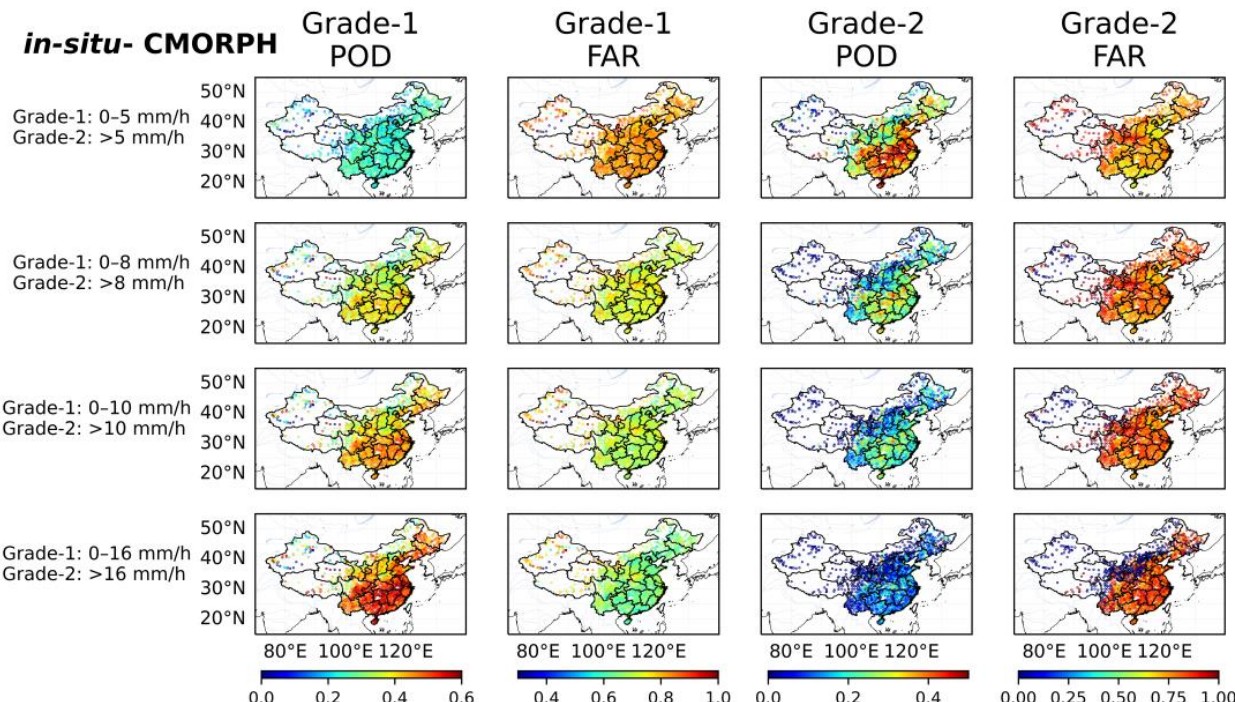

**Figure 11.** POD and FAR of the CMORPH data under different classification thresholds. The first and second columns are POD and FAR of Grade-1 precipitation, and the third and fourth columns are POD and FAR of Grade-2 precipitation. The standards corresponding to each line are displayed on the left side of the line.

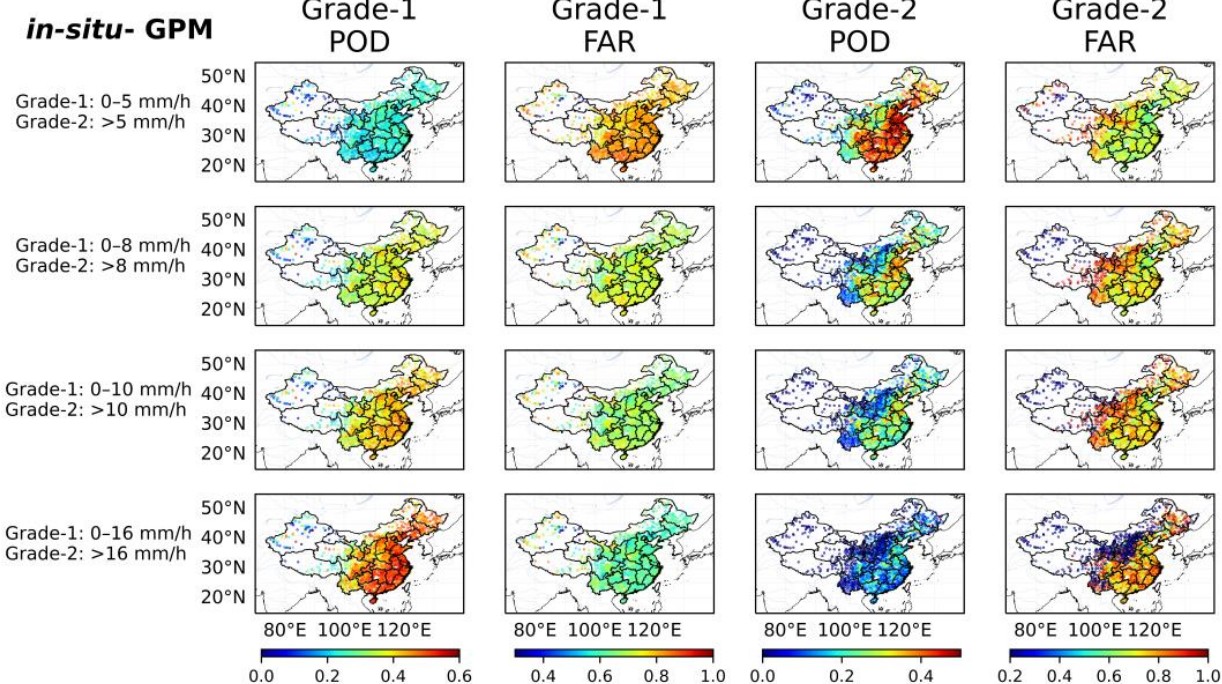

**Figure 12.** Same as Figure 11, but for the GPM data.

## 4. Conclusions

In this study, the overall and seasonal average performances of two different global satellite QPE data sets were compared and validated based on the *in-situ* precipitation data obtained from 2167 rain gauge stations in China from 2016 to 2020. Furthermore,

the possible influences of the errors on determining the precipitation intensity grade were also discussed. This study proposed that when using global gridded satellite QPE data for climate or meteorological studies, especially when only the precipitation intensity grade is used, we should carefully consider the accuracy of the data and even make some corrections. The main conclusions are as follows.

Compared with the CMORPH data, the errors of the GPM data were relatively small. Moreover, the errors of the two gridded satellite QPE data (GPM and CMORPH) showed some seasonal variation characteristics. For these two QPE data sets, the high-error area was the most widely distributed in summer, covering eastern and southern China, while the high-error areas in spring and autumn were mainly in Guangdong Province and Hainan Province. In addition to northern China in winter, a strong correlation was found between the GPM data and *in-situ* data at other times and in other regions. However, the correlation between the CMORPH data and *in-situ* data was poor, which may be because it is not adjusted with seasonal GPCP SG surface precipitation data. In addition, the production of the CMORPH data without the use of DPR data and algorithms that take into account more factors (such as the vertical structure of the atmosphere and the type of precipitation) also contributed to the poor performance of the CMORPH data compared to GPM data.

When discussing the influence of the error between satellite QPE data and the *in-situ* precipitation data on the accuracy of classification, we were surprised to find that with the increase in precipitation intensity, the related satellite QPE data often obviously underestimated extreme precipitation (both the GPM data with good performance in error analysis and CMORPH data with poor performance in error analysis), although they tended to overestimate precipitation according to the statistics of all precipitation greater than 0 mm/h from 2016 to 2020 (possibly affected by a large number of weak precipitation cases). This led to the lower accuracy of the satellite QPE data judgment for precipitation with a higher intensity grade. When the threshold value of heavy precipitation was reduced, the accuracy of the satellite QPE data division increased, but the accuracy of the satellite QPE data of moderate precipitation decreased. Therefore, considering the different thresholds used in this study (the definition of heavy rain in China was 8.1–16 mm/h, but the accuracy of the satellite QPE data when using 8.1 mm/h as the threshold was very different from that when using 16 mm/h), it is necessary to be aware of the great impact of the errors of these gridded precipitation products on the experiments over China, particularly in determining precipitation intensity grades. Using advanced scientific data matching methods and incorporating more types of high-resolution QPE data can improve the credibility of the conclusion regarding precipitation intensity classification. Further research can explore these avenues to make the findings more robust and scientifically sound.

**Author Contributions:** Conceptualization, X.W. and Z.L.; methodology, X.W.; software, X.W; validation, Y.Y.; formal analysis, X.W.; investigation, X.W. and Y.Y.; resources, Y.Y. and B.L.; data curation, Y.Y.; writing—original draft preparation, X.W.; writing—review and editing, Y.Y. and B.L.; visualization, B.L.; supervision, B.L.; project administration, B.L.; funding acquisition, B.L. All authors have read and agreed to the published version of the manuscript.

**Funding:** This research was funded by the National Natural Science Foundation of China (Grant No. 41975020).

**Data Availability Statement:** Publicly available data sets were analyzed in this study. These data can be found here: (1) CMORPH data: https://www.ncei.noaa.gov/data/cmorph-high-resolution-global-precipitation-estimates/access/30min/8km/ (accessed on 3 March 2019). (2) GPM data: https://gpm1.gesdisc.eosdis.nasa.gov/data/GPM_L3/GPM_3IMERGHH.06/ (accessed on 18 June 2019).

**Conflicts of Interest:** The authors declare no conflict of interest.

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
