# Peer review of "Representativeness of Two Global Gridded Precipitation Data Sets in the Intensity of Surface Short-Term Precipitation over China"

_remotesensing, doi:10.3390/rs15071856_

Round 1
Reviewer 1 Report (Previous Reviewer 1)
The authors have improved the paper according to the reviewers comments and in my opinion the current version of the manuscript can be considered for publication.
Author Response
Please see the attachment.

Reviewer 2 Report (Previous Reviewer 2)
The authors have responded to my questions/concerns. The work can be accepted for publication.
Author Response
Please see the attachment.

Reviewer 3 Report (New Reviewer)
General comments
In the manuscript “Representativeness of two global gridded precipitation datasets in the intensity of surface short-term precipitation over China”, the performance of two sets of global satellite precipitation gridded data on the simulated intensity of hourly precipitation in China was evaluated and compared. The results and variations of the modelling for different levels of intensity of hourly rainfall are examined in detail. This work followed a standard framework of evaluation of gridded meteorological data. The overall writing and structure of the manuscript is well organized. The line of reasoning is acceptable, the methods are appropriate, and the results are clearly presented. However, there are the following issues that need further explanation by the authors.
Specific comments
1. In the introduction section, the authors need to present more clearly the shortcomings of the existing studies on satellite precipitation evaluation and highlight the advancement of this study compared to the existing studies on satellite precipitation evaluation. For example, is there a difference in the perspective of comparison or of considering the intensity of hourly precipitation? In addition, the authors should add more recent studies on the evaluation of satellite precipitation products to more adequately highlight the progress of this manuscript compared to these studies. Including but not limited to the following studies. The authors should identify points of progress by comparison with these papers in the introduction or discussion.
Hosseini-Moghari, S. M., Sun, S., Tang, Q., & Groisman, P. Y. (2022). Scaling of precipitation extremes with temperature in China’s mainland: Evaluation of satellite precipitation data. Journal of Hydrology, 606, 127391.
Lei, H., Zhao, H., & Ao, T. (2022). Ground validation and error decomposition for six state-of-the-art satellite precipitation products over mainland China. Atmospheric Research, 269, 106017.
Zhang, Y., Wu, C., Yeh, P. J. F., Li, J., Hu, B. X., Feng, P., & Lei, Y. (2022). Evaluation of multi-satellite precipitation products in estimating precipitation extremes over mainland China at annual, seasonal and monthly scales. Atmospheric Research, 279, 106387.
Yu, C., Hu, D., Liu, M., Wang, S., & Di, Y. (2020). Spatio-temporal accuracy evaluation of three high-resolution satellite precipitation products in China area. Atmospheric research, 241, 104952.
Peng, F., Zhao, S., Chen, C., Cong, D., Wang, Y., & Ouyang, H. (2020). Evaluation and comparison of the precipitation detection ability of multiple satellite products in a typical agriculture area of China. Atmospheric Research, 236, 104814.
Ma, Q., Li, Y., Feng, H., Yu, Q., Zou, Y., Liu, F., & Pulatov, B. (2021). Performance evaluation and correction of precipitation data using the 20-year IMERG and TMPA precipitation products in diverse subregions of China. Atmospheric Research, 249, 105304.
2. The available satellite precipitation products are extensive and the authors needed to provide an additional explanation as to WHY only two of them were chosen for comparison and evaluation in the study? Higher resolution? Longer time scale? Or maybe some other reason.
3. What is the time period of this study? The authors evaluate hourly precipitation intensity and frequency of the different intensities on a multi-year average? Which years specifically? Only the number of days with precipitation was considered? All this information should be presented more clearly to the reader.
4. Does the alignment of the satellite precipitation gridded data with the in-situ data have an influence on the accuracy of the evaluation results? The authors briefly present the method of matching in lines 177-181, I would suggest that the authors could go into more detail and consider whether the matching of the stations to a particular satellite precipitation grid cell or to several cells around it would have an influence on the evaluation results?
5. The authors examine the differences in the modelling of hourly precipitation at different intensities, which is interesting. Could possible reasons for this difference be added to the discussion section? For instance, differences in the way or rationale for inverting precipitation from different satellite precipitation products?
Minor points
1. Line 23 and 24 of the Abstract are duplicated.
2. The latitude and longitude of Figs. 1 and 2 are too close to the edges of the map and it is suggested to adjust them.
Author Response
Please see the attachment.
Reviewer 4 Report (New Reviewer)
Reviewer’s Report on the manuscript entitled:
Representativeness of two global gridded precipitation datasets in the intensity of surface short-term precipitation over China
The authors compared two satellite precipitation data (CMORPH and GPM) with the hourly in-situ precipitation from China national surface stations from 2016 to 2020. Though the topic and results are generally interesting, the presentation and literature review must be improved. Therefore, I recommend major revisions. Please see below my comments.
General comments:
Literature review can be improved:
Line 61. TRMM data have been employed at a global scale in the following research that can be included here:
https://doi.org/10.3390/rs14215433
Line 68. GPM has been utilized in several studies due to its higher accuracy. Some of the recent articles that describe and utilize GMP can be included here, such as
https://doi.org/10.1016/j.jag.2023.103241
Daily precipitation GPM has also been assessed for arid regions in: https://doi.org/10.3390/rs11232840
Line 398. The standard errors of GPM (and CMORPH) are also provided by NASA. Did you verify whether the extreme precipitations have higher errors or not? Having that said, Figure 10 can include the error bars that comes with these products.
There are many typos/grammar/punctuation issues that must be fixed.
Quality of figures can be improved.
Specific comments:
Lines 17, What is QPE? All the abbreviations must be defined the first time they appear in the manuscript.
Line 22-23. Grammar issue. These sentences don’t have any verb and so they are incorrect.
Lines 24-25. Bad English. Please rewrite.
Lines 40-43. Other works on what subject is established? Please clarify.
Figures 1 and 2. The coordinated are too close to the box please give some space between them like Figures 3 and 4.
Figure 2. The country borders are not fully drawn. Please check and regenerate the panels. Similarly for Figures 3-9. I suggest to simply keep the China’s border and remove the borders of other countries entirely. See for example the second reference I suggested above as a guide.
Figures 4,5,6,7. Please move them in the appendix or supplementary materials. Because putting these figures in the body of the manuscript makes it hard for following the text and getting the message of this research.
An extra comma after “sat” in Equation 4.
Equation 6. Just write the numerator using square.
Line 232. “data” is plural. Here should be “…data have been” similarly in line 281, etc. Please check and correct elsewhere.
Line 337. Figure 10. As far as I can see from Figure 10 both QPE values are less than the in-situ weak and heavy precipitation. Please check. Perhaps you meant overestimate the weak and underestimate the heavy? If so, it can create confusion. Please rephrase.
Line 385. I know you wanted to emphasize this, but a paragraph should have at least two sentences. Either use numbering/bullet points or merge it to the following paragraph.
Please also add the limitations of your work in the conclusion section
Not all the references follow the MDPI guidelines. Please check and correct
Thank you!
Regards,
Round 2
Reviewer 3 Report (New Reviewer)
Thanks to the author for the changes, the comments were well addressed.
Author Response
We appreciate your feedback and hope that you continue to enjoy the content of the revised article.
Reviewer 4 Report (New Reviewer)
Dear authors,
Thank you for addressing most of my comments. Please see below my remaining comments.
Line 18. Please remove "seriously"
Line 26. Please replace "seriously" with "significantly".
Some of the articles that I suggested were not included. For example, the paper discussed the application of GPM in environmental monitoring in Italy could also be included in line 62. This will expand your literature review further and not just be limited to China.
Line 266. This sentence is ambiguous. Please rewrite.
Please check the reference numbers in the text that should match the references in the References.
Reference 5. Please check. You wrote doi twice. Please check all the references and ensure they have the same format/style as suggested by MDPI.
Thank you!
Author Response
Please see the attachment.

This manuscript is a resubmission of an earlier submission. The following is a list of the peer review reports and author responses from that submission.
Round 1
Reviewer 1 Report
Review of “Representativeness of two global gridded precipitation datasets in the intensity of surface precipitation over China” by Xiaocheng Wei, Yu Yu, Bo Li, Zijing Liu.
The main purpose of this paper is to assess the suitability over China of two global precipitation datasets based on satellite data, i.e. data from the Climate Prediction Center morphing (CMORPH) and data from the Global Precipitation Measurement (GPM) mission. To this aim, the authors compare these two satellite precipitation data with the hourly in-situ precipitation data from China weather stations both in terms of quantitative precipitation estimate (QPE) and in terms of classification of precipitation intensity.
General Comment
The authors have improved the paper according to the reviewers comments and in my opinion the current version of the manuscript could be considered to be published with minor changes.
Specific Comments
Introduction. Page 2. Line 84: “…probability of detection (HR)…”
The acronym HR is not correct for probability of detection, please fix it.
Data and Methods. Page 3. Line 135.
There is a string in Chinese, please remove it.
Figures 8, 9 and 10 report the strings POD and FAR instead of HR and MR, please correct them.
Figure 11 report “in-suit” in the legend, please correct it.
Discussion and Conclusions. Page 17.
In my first review I raised a question related to the apparent contradictory conclusions in the paper. The answer of the authors is convincing, but the conclusions of the paper have not changed compared to its old version. I suggest to clarify the correct interpretation of the results also in conclusions, emphasizing the considerations made in the manuscript (lines 344-348).
Author Response
Thank you for your suggestion. Please see the attachment.

Reviewer 2 Report
The authors have answered my questions. However, I'm still not convinced if two years of data is enough for this kind of analysis. Obviously, climate data has trends that are not properly represented in two years of data. In addition, the QPE approach investigated in this study is very commonly used in many other precipitation analysis papers. So, I see no value added to the literature. I leave the decision to the editor whether to accept or reject the paper.
Author Response
Thank you for your work. Please see the attachment.

Reviewer 3 Report
1. Authors should explain the novelty of this study for the readers to catch the importance of the study over other previous studies.
2. One major issue of the introduction section is that it does not establish the research gap before discussing the aim of the study in lines 115-118.
3. More details are needed about the rationale of choosing GPM and CMORPH precipitation data over others.
4. Authors should revise the manuscript thoroughly to avoid typographical/spelling errors, sentence structure issues and other formatting issues.
5. Conclusions should be rewritten to include key findings without repeating the results.
6. Some sentences are too general to be of much use, for example, in Line 370-373, Line 344-346, Line 317-320, more words and discussions are needed.
7. Lines 200-213 do not belong to Data and methods, Lines 215-222 are suggested to put in Methods.
8. More words are needed why the four stations are selected in line 335.
Author Response

(The authors gave the same response as above.)

Round 2
Reviewer 3 Report
no much improvements has been made.
